# Further Insights into the Architecture of the *P_N_* Promoter That Controls the Expression of the *bzd* Genes in *Azoarcus*

**DOI:** 10.3390/genes10070489

**Published:** 2019-06-27

**Authors:** Gonzalo Durante-Rodríguez, Paloma Gutiérrez-del-Arroyo, Marisela Vélez, Eduardo Díaz, Manuel Carmona

**Affiliations:** 1Microbial and Plant Biotechnology Department, Centro de Investigaciones Biológicas-CSIC, Ramiro de Maeztu, 9, 28040 Madrid, Spain; 2Biocatalysis Department, Institute of Catalysis and Petrochemistry-CSIC, Marie Curie, 2, Cantoblanco, 28049 Madrid, Spain

**Keywords:** regulation, benzoate, anaerobic, *Azoarcus*, promoter architecture

## Abstract

The anaerobic degradation of benzoate in bacteria involves the benzoyl-CoA central pathway. *Azoarcus*/*Aromatoleum* strains are a major group of anaerobic benzoate degraders, and the transcriptional regulation of the *bzd* genes was extensively studied in *Azoarcus* sp. CIB. In this work, we show that the *bzdR* regulatory gene and the *P_N_* promoter can also be identified upstream of the catabolic *bzd* operon in all benzoate-degrader *Azoarcus*/*Aromatoleum* strains whose genome sequences are currently available. All the *P_N_* promoters from *Azoarcus*/*Aromatoleum* strains described here show a conserved architecture including three operator regions (ORs), i.e., OR1 to OR3, for binding to the BzdR transcriptional repressor. Here, we demonstrate that, whereas OR1 is sufficient for the BzdR-mediated repression of the *P_N_* promoter, the presence of OR2 and OR3 is required for de-repression promoted by the benzoyl-CoA inducer molecule. Our results reveal that BzdR binds to the *P_N_* promoter in the form of four dimers, two of them binding to OR1. The BzdR/*P_N_* complex formed induces a DNA loop that wraps around the BzdR dimers and generates a superstructure that was observed by atomic force microscopy. This work provides further insights into the existence of a conserved BzdR-dependent mechanism to control the expression of the *bzd* genes in *Azoarcus* strains.

## 1. Introduction

Aromatic compounds are the most widespread organic compounds in nature after carbohydrates. Moreover, the release of aromatic compounds into the biosphere increased considerably over the last century as a consequence of industrial activity [1]. Many of these compounds are toxic and/or carcinogenic, thus representing major persistent environmental pollutants. Some specialized microorganisms, mainly bacteria and fungi, adapted to degrade a wide variety of aromatic compounds aerobically and/or anaerobically [2]. Whereas the aerobic degradation of aromatics is extensively studied [3,4,5], the anaerobic degradation pathways are much less well studied, especially regarding the genes involved in these anaerobic processes [6,7]. However, many environments are anoxic and, thus, the anaerobic degradation of aromatic compounds has great importance at the ecological level [6,7,8]. The anaerobic degradation of a wide variety of aromatic compounds converges into a few central pathways (catabolic funnel) that carry out the reductive de-aromatization and further conversion of the intermediates to the central metabolism. Most monocyclic aromatic compounds are channeled and activated to the central intermediate benzoyl-CoA [2,8,9]. Benzoate was used over the two last decades as a model compound to study the main pathway for anaerobic degradation of aromatic compounds, i.e., the central route of benzoyl-CoA [7,9]. The anaerobic degradation of benzoate takes place through a peripheral route consisting of a single reaction that activates benzoate to benzoyl-CoA. Benzoyl-CoA is then de-aromatized by the action of a reductase, the only oxygen-sensitive enzyme within the benzoyl-CoA pathway, which generates cyclohexa-1,5-diene-1-carbonyl-CoA, which is further converted into 3-hydroxypimelyl-CoA through a modified β-oxidation mechanism (Figure 1A) [2,7,8,9,10]. The genes encoding the enzymes involved in the anaerobic degradation of benzoate were reported in some phototrophic bacteria, e.g., *Rhodopseudomonas palustris* strains (*bad* genes), facultative anaerobes, e.g., *Thauera*, *Azoarcus*, *Aromatoleum*, *Magnetospirillum*, and *Sedimenticola* strains (*bcr*/*bzd* genes), and in strict anaerobes, e.g., *Geobacter*, *Syntrophus*, and *Desulfobacula* strains (*bam* genes) [8,9,10,11,12,13,14,15]. The expression of the catabolic genes is controlled by specific transcriptional regulators, but only a few such regulators are described so far, mainly the BadM protein in *R. palustris* [16,17], the BamVW and BgeR proteins in *Geobacter* strains [18,19], and the BzdR protein in *Azoarcus* sp. CIB [20,21].

Among facultative anaerobes, bacteria of the *Azoarcus* genus, recently re-classified within the new *Aromatoleum* genus [22], represent one of the major groups of aromatic degraders. *Azoarcus* sp. CIB is a denitrifying β-Proteobacterium able to aerobically and/or anaerobically degrade a wide variety of aromatic compounds, most of which are channeled to the benzoyl-CoA central pathway encoded by the *bzd* genes. The *bzd* genes from strain CIB are clustered together in a large catabolic operon (*bzdNOPQMSTUVWXYZA*) driven by the *P_N_* promoter (Figure 1B). The specific transcriptional regulation of the *bzd* operon is carried out by the BzdR repressor encoded immediately upstream of the catabolic operon (Figure 1B) [20,23]. Benzoyl-CoA, the first intermediate during the anaerobic degradation of benzoate, is specifically recognized by BzdR and acts as the inducer molecule allowing the expression of the *bzd* catabolic genes [23]. BzdR binds to the *P_N_* promoter at three different regions that contain a direct repetition of the sequence TGCA [20]. In addition to BzdR, two additional regulatory proteins account for an overimposed regulation of the *P_N_* promoter. AcpR, a regulator belonging to the well-known fumarate and nitrate reductase (FNR)/cyclic AMP receptor protein (CRP) superfamily of transcriptional regulators, is essential for the activation of the *P_N_* promoter under oxygen deprivation conditions [24]. Furthermore, the AccR response regulator interacts with *P_N_* and inhibits its activity in response to several organic acids such as succinate, malate, or acetate, accounting for the carbon catabolite control of the *bzd* genes [23,25].

In this work, we expand the current knowledge on the organization of the *bzd* clusters in all *Azoarcus*/*Aromatoleum* strains sequenced so far that are able to degrade benzoate anaerobically. The evolutionary conservation of a common *P_N_* promoter architecture with three BzdR operator regions can be highlighted. The role of the three BzdR operator regions was studied, and the formation of a BzdR/*P_N_* superstructure was confirmed and visualized.

## 2. Material and Methods

### 2.1. Bacterial Strains, Plasmids, and Growth Conditions

Bacterial strains and plasmids used in this study are listed in Table 1.

*Escherichia coli* cells were grown at 37 °C in lysogeny broth (LB) medium [34]. When required, *E. coli* cells were grown anaerobically at 30 °C in LB medium supplemented with 0.1% casamino acids [34] and 10 mM KNO_3_ as a terminal electron acceptor. *Azoarcus* strains were anaerobically grown at 30 °C in MC medium as described previously [23]. Where appropriate, antibiotics were added at the following concentrations: ampicillin, 100 µg∙mL^−1^; chloramphenicol, 30 µg∙mL^−1^; gentamicin, 7.5 µg∙mL^−1^; and kanamycin, 50 µg∙mL^−1^.

### 2.2. Molecular Biology Techniques

Recombinant DNA techniques were carried out according to published methods [26]. Plasmid DNA was prepared with a High Pure Plasmid Isolation Kit (Roche Applied Science, Penzberg, Germany). The DNA fragments were purified with Gene Clean Turbo (Q-BIOgene, Carlsbad, CA, USA). Oligonucleotides were supplied by Sigma (St. Louis, MO, USA). The sequence of inserts/DNA fragments was confirmed by DNA sequencing with an ABI Prism 377 automated DNA sequencer (Applied Biosystems, Foster City, CA, USA). Transformation of *E. coli* was carried out by using competent cells prepared by the RbCl method [34] or by electroporation (Gene Pulser, Bio-Rad, Cambridge, MA, USA) [34]. Plasmids were transferred from *E. coli* S17-1λ*pir* (donor strain) to *Azoarcus* sp. CIB recipient strains via biparental filter mating as previously reported [23]. Proteins were analyzed by SDS-PAGE as described previously [26].

### 2.3. Sequence Data Analyses

For bioinformatic inspection of genes and regulatory regions of interest, we employed the BioEdit Sequence Alignment Editor [35]. The Basic Local Alignment Search Tool (BLAST) platform [36] was used for studying the similarity/identity of proteins encoded in the *bzd* cluster. The amino-acid sequences of the open reading frames of proteins encoded in the *bzd* operon were compared with those present in databases using the TBLASTN algorithm at the National Center for Biotechnology Information (NCBI) server [37] (http://blast.ncbi.nlm.nih.gov/Blast.cgi). Nucleotide and protein alignments were done with ALIGN [38] and CLUSTALW [39], respectively, in the BioEdit editor.

### 2.4. Overproduction and Purification of His_6_-BzdR, His_6_-NBzdRL, and His_6_-FNR* Proteins

Plasmid pQE32-His_6_BzdR and pQE32-His_6_NBzdRL produce N-terminally His_6_-tagged BzdR and NBzdRL proteins (N-terminal domain plus the BzdR linker), respectively (Table 1). Plasmid pQE60-His_6_FNR* produces a C-terminally His_6_-tagged Fnr* (Table 1). All these plasmids express the cloned genes under control of the *P_T5_* promoter and two *lac* operator boxes. The His_6_-tagged proteins were overproduced in *E. coli* M15 strain also harboring plasmid pREP4, and they were purified following protocols previously established [20].

### 2.5. Analytical Ultracentrifugation Assays

To perform experiments of sedimentation equilibrium of His_6_-NBzdRL protein bound to DNA, different concentrations of this protein (0.5 to 10 μM) were incubated with the DNA fragment (0.05 to 0.2 μM). The DNA fragments used were the *P_N_I* promoter (145 bp) and the complete *P_N_* promoter (*P_N_*, 253 bp) that were obtained by PCR using the oligonucleotide pairs BLINE (5’–CGTGCCTGACATTTGACTTAGATC–3’) and 3IVTPN (5’–CGGGAATTCCATCGAACTATCTCCTCTGATG–3’, *Eco*RI site underlined), and RET1 (5’–CCGAGCCTCGCGTTTTACTGC–3’) and 3IVTPN, respectively. The ultracentrifugation experiments were conducted in a buffer containing 50 mM NaH_2_PO_4_, 300 mM KCl, 100 mM imidazole, pH 8, as previously described [21]. Since the BzdR protein aggregates in experiments of sedimentation equilibrium at concentrations higher than 2 μM, we used the NBzdRL protein that is totally soluble at 10 μM. The presence of DNA was checked by measuring the absorbance at 260 nm. The distribution of sediment coefficients obtained in the sedimentation velocity experiments was determined using the SEDFIT program as reported previously [21].

### 2.6. Construction of P_N_I::lacZ and P_N_II::lacZ Translational Fusions and β-Galactosidase Assays

The *P_N_::lacZ* translational fusion was described previously [23]. To generate two truncated versions of the *P_N_* promoter, *P_N_I* and *P_N_II*, fused to the *lacZ* reporter gene, we firstly PCR-amplified the truncated promoters. The oligonucleotides used for amplification of the *P_N_I* promoter were BKIN (5’–GGGGTACCCGTGCCTGACATTTGACTTAGATC–3’; *Kpn*I site underlined) and 5BZN (5’–GCTCTAGACCCATCGAACTATCTCCTCTGATG–3’; *Xba*I site underlined). For the amplification of the *P_N_II* promoter, the oligonucleotides used were NIU PN-II (5´–GGGGTACCCAAGAAAGATTGCAGTTTTCCATG; *Kpn*I site underlined) and 5BZN. The template used for amplification of the promoter regions was the pECOR7 plasmid (Table 1). After PCR-amplification, both DNA fragments were digested with *Xba*I and *Kpn*I restriction enzymes, and later cloned in plasmid pSJ3 in-frame with *lacZ* gene, thus generating the corresponding *P_N_I::lacZ* and *P_N_II::lacZ* translational fusions in plasmids pSJ3-P_N_I and pSJ3-P_N_II, respectively. These translational fusions were subsequently subcloned by *Eco*RI/*Hind*III double digestion into plasmid pBBR1MCS-5 generating plasmids, pBBR5-P_N_I and pBBR5-P_N_II, respectively. For determination of promoter activity, pSJ3-P_N_I and pSJ3-P_N_II were transformed in *E. coli* MC4100, while pBBR5-P_N_I and pBBR5-P_N_II were transferred to *Azoarcus* sp. CIB [23]. *E. coli* MC4100 carrying the *P_N_::lacZ*, *P_N_I::lacZ*, or *P_N_II::lacZ* fusions was grown anaerobically in LB-rich medium, whereas *Azoarcus* sp. CIB carrying the *P_N_::lacZ*, *P_N_I::lacZ*, or *P_N_II::lacZ* fusions was grown anaerobically in MC medium in the presence of 3 mM benzoate or 0.2% (*w*/*v*) succinate. Cultures were incubated at 30 °C until the mid-exponential phase. The β-galactosidase activity (in Miller Units) was determined in permeabilized cells (using 0.1% SDS and chloroform) according to the method described by Miller [34].

### 2.7. In Vitro Transcription Assays

In vitro transcription assays were performed as previously published [40,41] using plasmids pJCD-P_N_, pJCD-P_N_I, and pJCD-P_N_II (0.5 nM) as supercoiled templates. To construct these plasmids, firstly, 585-bp, 139-bp, and 209-bp DNA fragments containing the *P_N_*, *P_N_I*, and *P_N_II* promoters, respectively, were PCR-amplified from the *Azoarcus* sp. strain CIB chromosome. The forward and reverse oligonucleotides 5IVTPN (5’–CGGAATTCCGTGCATCAATGATCCGGCAAG–3’; *Eco*RI site underlined) and 3IVTPN (5’–CGGAATTCCATCGAACTATCTCCTCTGATG–3’; *Eco*RI site underlined), BIN (5’–CGGAATTCCGTGCCTGACATTTGACTTAGATC–3’; *Eco*RI site underlined) and 3IVTPN, and BIIN (5’–CGGAATTCCAAGAAAGATTGCAGTTTTCCATG–3’; *Eco*RI site underlined) and 3IVTPN, respectively, were used for PCR-amplification. Then, the DNA fragments were *Eco*RI restricted, and cloned into the *Eco*RI-restricted pJCD01 cloning vector, giving rise to plasmids pJCD-P_N_, pJCD-P_N_I, and pJCD-P_N_II, respectively (Table 1). Reactions (50-µL mixtures) were performed in a buffer containing 50 mM Tris-HCl (pH 7.5), 50 mM KCl, 10 mM MgCl_2_, 0.1 mM bovine serum albumin, 10 mM dithiothreitol, and 1 mM ethylenediaminetetraacetic acid (EDTA). Unless otherwise indicated, each DNA template was premixed with 100 nM σ^70^-containing *E. coli* RNA polymerase (RNAP) holoenzyme (Amersham plc, Amersham, UK) and different amounts of purified His_6_-Fnr*. For multiple-round assays, transcription was then initiated by adding a mixture of 500 μM (each) adenine, cytosine, and guanine triphosphate (ATP, CTP, and GTP); 50 μM uridine triphosphate (UTP); and 2.5 μCi of [α^32^P]-UTP (3000 Ci∙mmol^−1^). After incubation for 15 min at 37 °C, the reactions were stopped with an equal volume of a solution containing 50 mM EDTA, 350 mM NaCl, and 0.5 mg of carrier transfer RNA (tRNA; yeast tRNA from Thermo Fisher Scientific, Whatham, MA, USA) per ml. The messenger RNA (mRNA) produced was then precipitated with ethanol, separated on a denaturing 7 M urea/4% polyacrylamide gel, and visualized by autoradiography. Transcript levels were quantified with a Bio-Rad Molecular Imager FX system and using the ImageJ software [42].

### 2.8. Gel Retardation Assays

The *P_N_I* DNA probe (145 bp) was obtained as described previously [21], digested with *Eco*RI, and labeled by filling in the overhanging *Eco*RI-digested end with [α-^32^P]dATP (6000 Ci∙mmol^−1^; Amersham Biosciences, Amersham, UK) and the Klenow fragment of *E. coli* DNA polymerase as described previously [20]. The DNA probe was mixed with the purified proteins at the concentration indicated in each assay. The retardation reaction mixtures contained 20 mM Tris-HCl pH 7.5, 10% glycerol, 2 mM β-mercaptoethanol, 50 mM KCl, 0.05 nM DNA probe, 250 μg∙mL^−1^ bovine serum albumin, and purified His_6_-BzdR protein in a 9-μL final volume. The samples were fractionated by electrophoresis in 5% polyacrylamide gels buffered with 0.5× TBE (45 mM Tris borate, 1 mM EDTA). The gels were dried onto Whatman 3MM paper and exposed to Hyperfilm MP (Amersham Biosciences, Little Chalfont, UK).

### 2.9. Analysis of Protein–DNA Interaction by Atomic Force Microscopy

To perform atomic force microscopy (AFM) experiments, a DNA fragment of 1225 bp, termed P_N_L, from position −505 to +720 and covering the intergenic region between *bzdR* and *bzdN* genes, as well as the 5´-end of the *bzdN* gene, was PCR-amplified using as template the plasmid pECOR7, and the oligonucleotides 3REG (5’–GGGGTACCCGTGCATCAATGATCCGGCAAG–3’; *Kpn*I site underlined) and N3 (5’–TTCAGCATCTCGTTGTGCTC–3’). The purified His_6_-BzdR protein was obtained as detailed above. The binding reactions were performed for 15 min at room temperature using retardation buffer (20 mM Tris·HCl, pH 7.5, 10% glycerol, 2 mM β-mercaptoethanol, 50 mM KCl). The protein and DNA concentrations were optimized to ensure that the unbound protein (probably adsorbed to the surface) did not disturb the image. The incubation of the complexes was performed at a final concentration of 2 nM P_N_L fragment with [P_N_L]:[BzdR] ratios of 1:28 and 1:70. The samples of complexes were diluted 10 times in retardation buffer, in the presence of MgCl_2_ or NiCl_2_ at concentrations between 3 and 10 mM. Each reaction mixture was then deposited on a freshly exfoliated mica surface (which provides a clean and atomically flat substrate). After 1 min of incubation on the mica, the non-adsorbed material was removed by successive washes with the same adsorption buffer. Finally, the samples, imaged while immersed in buffer, were visualized with an atomic force microscope from Nanotec Electronica S.L (Tres Cantos, Spain) operated in jump mode [43]; the images were processed with the WSxM program [44].

## 3. Results and Discussion

### 3.1. The Genetic Organization of the *bzd* Cluster and the Architecture of the *P_N_* Promoter Are Conserved in the *Azoarcus*/*Aromatoleum* Genus

The anaerobic degradation of benzoate is a common feature among many strains of the *Azoarcus*/*Aromatoleum* genus [22]. By mining in the available genome sequences of *Azoarcus* aromatic degrader strains, i.e., those of *A. tolulyticus*, *A. toluclasticus*, *Azoarcus* sp. KH32C, and *Azoarcus* sp. PA01, we found the presence of genes homologous to the *bzd* genes responsible of the anaerobic degradation of benzoate in *A. evansii*, *Azoarcus* sp. CIB, and *A. aromaticum* EbN1 [8,45]. Interestingly, in all these *Azoarcus* strains, the *bzd* genes are arranged in a cluster that contains a *bzdR* regulatory gene located upstream of a putative *bzd* catabolic operon. The *bzd* operon is organized into at least three different functional modules: (i) the activation module (*bzdA*) that codes for the enzyme that generates benzoyl-CoA; (ii) the de-aromatization module (*bzdNOPQMV*) that encodes the benzoyl-CoA reductase and auxiliary enzymes; and (iii) the modified β-oxidation module (*bzdWXY*) that encodes the enzymes that generate 3-hydroxypimelyl-CoA (Figure 1) [8,10]. A close inspection of the promoter region of the predicted *bzd* operons from all *Azoarcus* strains revealed the conservation of the *P_N_* promoter described in *Azoarcus* sp. CIB [20] (Figure 2). Remarkably, it should be noted that this organization found in *Azoarcus*/*Aromatoleum* was not observed in closely related aromatic-compound-degrading denitrifying bacteria such as those of the genus *Thauera*.

As described before, the *P_N_* promoter of strain CIB shows the transcription start site located minus 75 nucleotides with respect to the ATG start codon of the *bzdN* gene, and the −10 and −35 boxes of interaction with the σ^70^ subunit of the RNAP were also identified [20] (Figure 2). Three BzdR operator regions were detected in the *P_N_* promoter by DNase I footprinting, i.e., operator region I (OR1), located between positions −32 and +31 (with respect to the transcription start site), operator region II (OR2) (−83 to −63), and operator region III (OR3) (−146 to −126) [20]. The operator region recognized by the AcpR activator protein overlaps the −35 box of RNAP [24] (green box, Figure 2). Interestingly, the *P_N_* promoter sequences of all *bzd* operons from *Azoarcus* strains exhibited the three BzdR operator regions (OR1, OR2, and OR3) and the operator region that binds to the AcpR transcriptional activator (Figure 2). The OR2 and OR3 regions of *P_N_* contain the conserved TGCA(N_6_)TGCA palindromic sequence where N_6_ is a six-nucleotide A-rich region in OR3. OR1 is longer and it contains two conserved palindromic sequences, TGCA(C)T(G/C)(C/G)A and TGCA(N_15_)TGCA, located between the −35 and −10 boxes and downstream of the transcription start site, respectively (Figure 2).

The TGCA palindromic structures were also described in other operator regions such as those recognized by the CopG repressor from *Streptococcus* strains, a 45 amino-acid homodimer encoded in the pMV158 plasmid family, at the *Pcr* promoter that drives the expression of the *copG* and *repB* genes involved in plasmid replication [46,47]. Another example of a regulatory protein that recognizes operator boxes that include the TGCA sequence is the XylS regulator, which controls the expression of the operon for *m*-xylene and toluene catabolism in *Pseudomonas putida*. XylS acts as an activator binding to a region of the *Pm* promoter that includes two imperfect palindromes of the TGCA(N_6_)GGNTA type [48]. The regulatory protein BenR, involved in the regulation of the aerobic catabolism of benzoate and other analogues in *P. putida*, also seems to recognize this type of imperfect palindrome [49].

The conservation of the three BzdR operator regions in all *P_N_* promoters identified so far suggests that they are required for the accurate regulation of this promoter. To further study the role of the three BzdR operator regions of *P_N_*, we performed in vivo and in vitro approaches with a set of truncated versions of this promoter.

### 3.2. In Vivo Studies on the Activity of Truncated P_N_ Promoters

As described previously, the activity of the *P_N_* promoter requires the RNAP and the interaction with the AcpR, a transcriptional activator from the well-known FNR/CRP superfamily [50] that binds at a consensus sequence centered at position −41.5 from the transcription start site and overlapping the RNAP −35 box [24] (Figure 2). We demonstrated previously that the FNR protein from *E. coli* is able to bind and activate the *P_N_* promoter [24]. On the other hand, despite the *P_N_* promoter containing three BzdR operators, the OR1 region itself overlaps the RNAP −10 box and +1 sites (Figure 2), thus having the requested sequence for the interaction with RNAP. Therefore, it could be expected that a truncated promoter just containing OR1 and the AcpR operator could behave as a functional BzdR-controlled minimal promoter. To check this assumption, we engineered a set of truncated *P_N_* promoters (Figure 3A) and compared their behavior with that of the complete *P_N_* promoter by in vivo assays. To this end, the truncated *P_N_I* promoter (spanning from position −61 to +79; carries OR1) and truncated *P_N_II* promoter (spanning from position −112 to +79; carries OR1 and OR2) (Figure 3A) were cloned in plasmid pSJ3, rendering plasmids pSJ3-P_N_I (contains the *P_N_I::lacZ* translational fusion) and pSJ3-P_N_II (contains the *P_N_II::lacZ* translational fusion) (Table 1). These plasmid constructions were independently transformed into *E. coli* MC4100 (pCK01) and *E. coli* MC4100 (pCK01-BzdR), respectively (Table 1). The β-galactosidase activity assays revealed that promoters *P_N_I* and *P_N_II* showed activity levels similar to those obtained with the complete *P_N_* promoter cloned in plasmid pSJ3P_N_ (Table 1) when present in the *E. coli* MC4100 (pCK01) strain (Figure 3B). Moreover, the BzdR protein was able to inhibit the activity of *P_N_I* and *P_N_II* as efficiently as in the case of the wild-type *P_N_* promoter in *E. coli* MC4100 (pCK01-BzdR) cells (Figure 3B). Therefore, these results show that *P_N_I* behaves as a functional minimal promoter that becomes fully active in the absence of BzdR and fully repressed in the presence of the BzdR protein.

To check the de-repression (induction) of the wild-type and truncated *P_N_* promoters by benzoyl-CoA, we expressed the *P_N_::lacZ*, *P_N_I::lacZ*, and *P_N_II::lacZ* fusions in *Azoarcus* sp. CIB. *Azoarcus* sp. CIB is able to produce benzoyl-CoA when benzoate is present in the culture medium. Benzoyl-CoA acts as inducer molecule allowing the de-repression of the system by inducing conformational changes in BzdR that lead to the release of the repressor from the promoter [21]. To perform the analysis in *Azoarcus*, the *lacZ* fusions were independently subcloned into the broad-host-range pBBR1MCS-5 vector generating plasmids pBBR5-*P_N_*, pBBR5-*P_N_I*, and pBBR5-*P_N_II*, respectively (Table 1). The plasmids were introduced by conjugation into *Azoarcus* sp. CIB, and the level of activity of each promoter was determined by β-galactosidase assays. As expected, the activity of the *P_N_*-derivative promoters expressed in *Azoarcus* cells grown in succinate (non-induction conditions) led to very low activity of all translational fusions (Figure 3C), revealing an efficient repression of the three *P_N_* promoters by the chromosomally encoded BzdR protein. When *Azoarcus* sp. CIB cells were grown in 3 mM benzoate (induction conditions), the complete *P_N_* promoter showed the expected activation (Figure 3C), thereby revealing the benzoyl-CoA-induced release of the BzdR repressor from the target promoter [20]. However, in contrast to the clear de-repression of the wild-type *P_N_* promoter in *Azoarcus* sp. CIB cells grown in benzoate, the *P_N_II* promoter showed a limited de-repression (Figure 3C), and almost no de-repression was observed with the truncated *P_N_I* promoter (Figure 3C).

In summary, the in vivo assays revealed that *P_N_I* is a fully active promoter in the absence of BzdR but a constitutively repressed promoter in the presence of BzdR. The OR2 and OR3 regions appear to be essential for the full de-repression (induction) of the *P_N_* promoter when benzoyl-CoA is produced in *Azoarcus* sp. CIB cells grown in benzoate. To further confirm the role of OR2 and OR3 in the de-repression of *P_N_* by benzoyl-CoA, several in vitro assays were performed.

### 3.3. Operator Regions II and III Are Needed for the Benzoyl-CoA-Dependent De-Repression of the *P_N_* Promoter

To further study the truncated *P_N_* promoters, we compared their activity levels to that of the wild-type *P_N_* promoter by performing in vitro transcription assays using the supercoiled plasmids pJCD-P_N_, pJCD-P_N_I, and pJCD-P_N_II (Table 1) as DNA templates, and the *E. coli* FNR* protein (a constitutively active AcpR ortholog [24]) as an activator. As shown in Figure 4, all three promoters generated the expected 184-nucleotide transcript in the absence of BzdR, but this transcript was lacking in the presence of the BzdR repressor. However, whereas a clear de-repression at the *P_N_* promoter was observed (about five times) when benzoyl-CoA was added to the BzdR-containing reaction, a slight de-repression (about two times) was observed with *P_N_II*, and benzoyl-CoA did not produce any de-repression effect at *P_N_I* (Figure 4). Thus, these results suggest that benzoyl-CoA was not able to alleviate the BzdR-mediated repression at *P_N_* when both OR2 and OR3 were missing. To check that the lack of de-repression at promoter *P_N_I* was a consequence of the promoter occupancy by BzdR even in the presence of the benzoyl-CoA inducer, gel shift assays were performed using the *P_N_I* probe in the presence of BzdR and increasing concentrations of benzoyl-CoA (Figure 5B). As expected, BzdR produced retardation of the *P_N_I* promoter probe, but increasing concentrations of benzoyl-CoA up to 2 mM, which released the BzdR repressor from the wild-type *P_N_* promoter (Figure 5A), were unable to induce the dissociation of the BzdR/*P_N_I* complex (Figure 5B). Since benzoyl-CoA promotes the release of BzdR from the *P_N_* promoter (Figure 5A) [20], this result suggests that the constitutive repression of *P_N_I* by BzdR is due to the irreversible binding of the repressor to the target promoter.

We demonstrated previously that the binding of BzdR to its cognate promoter is cooperative [21]. Sedimentation velocity analyses showed that 8–10 molecules of BzdR bind to the *P_N_* promoter, and it was postulated that each of the four TGCA palindromic regions bind a BzdR dimer, i.e., four BzdR molecules bound at OR1, two at OR2, and another two at OR3 [21]. Thus, it could be expected that the lack of de-repression at the *P_N_I* promoter in the presence of benzoyl-CoA inducer could be due to an excess of BzdR molecules bound to OR1 when both OR2 and OR3 are missing in the target promoter.

To check this hypothesis, we performed analytical ultracentrifugation experiments with the N-terminal domain of the BzdR protein (NBzdRL protein; residues 1–90) that is able to bind to *P_N_* [30]. Previous sedimentation velocity experiments revealed that NBzdRL was a dimer with S-values of 2.3 S, demonstrating that BzdR dimerization is an intrinsic property of the N-terminal domain [30]. The results presented here revealed that eight molecules of NBzdRL were bound to the wild-type *P_N_* promoter (Figure 6A), confirming the previous results obtained with the complete BzdR protein [30]. More relevant, the ultracentrifugation analyses showed that four molecules of NBzdRL were bound to the *P_N_I* fragment (Figure 6B). Hence, these results provide the first experimental evidence that two BzdR dimers bind to OR1 and support the previous assumption of the distribution of the eight BzdR molecules along the *P_N_* promoter [21].

### 3.4. Visualization of the BzdR/*P_N_* Complex

The results presented above indicate that three BzdR operator regions are conserved in all *P_N_* promoters identified so far, and they suggest that binding of four BzdR dimers to the wild-type *P_N_* promoter should lead to the formation of a big protein–DNA complex. Atomic force microscopy (AFM) permits direct visualization of the tridimensional structure of protein/DNA complexes arranged on an atomically flat surface and supplies data about the topology and stoichiometry of the complexes [51,52,53,54]. A 1225-bp DNA sequence (414 nm in length) spanning from position −505 to +720 with respect to the transcription start site was used as template (*P_N_L* fragment) to visualize the structure of *P_N_* interacting with BzdR when using a 50:1 [BzdR]:[DNA] ratio (Figure 7A). Images obtained when the three operator boxes were occupied by BzdR revealed a compact structure (Figure 7B). The lengths of the free (unoccupied) DNA that emerged at both sides of the compact structure were 108.9 ± 20 nm and 199.9 ± 44 nm (*n* = 16), respectively (Figure 7C, shown for one of the 16 samples). These sizes are slightly lower than those theoretically expected, i.e., 121 nm and 233 nm, respectively (Figure 7A). The total extension of the BzdR–DNA complex was reduced by 14 ± 4% with respect to the length of the naked *P_N_L* DNA fragment (Figure 7C).

These results are compatible with the formation of one loop of DNA around the BzdR proteins in the BzdR–DNA complex. The length of the DNA, including the DNA that is forming the loop, was estimated to be 398.6 ± 28.9 nm, which is compatible with the total length of the *P_N_L* DNA fragment (414 nm) and with the previous length calculated for the *P_N_L* naked template (400 ± 13 nm; [55]). Furthermore, the width of the BzdR–DNA “superstructure” had a value of 26.9 ± 6 nm (*n* = 10) (Figure 7D). Since a BzdR dimer resolved by electron microscopy is like a cylinder of 9 nm × 7 nm [21], the width of the superstructure observed by AFM is compatible with 4–5 dimers (8–10 molecules) of BzdR protein interacting with OR regions I, II, and III. These results are in good agreement with those obtained by analytical ultracentrifugation experiments (Figure 6A) [21]. In summary, the AFM results strongly suggest that the interaction of BzdR with the three operator regions of *P_N_* leads to the formation of a superstructure compatible with a DNA loop embracing 4–5 BzdR dimers by turning on itself (Figure 7). This type of superstructure was also described for other DNA–protein complexes such as that formed by the DNA/MutS protein [56], although other transcriptional regulators, such as the TodT protein that controls the aerobic toluene degradation pathway in *P. putida*, interact with the target DNA following a fork-type model [57]. The existence of a DNA loop wrapping the BzdR dimers as observed by AFM is also consistent with the DNase I footprinting studies that showed hyper-reactive nucleotides between OR1 and OR2 and between OR2 and OR3 [18], which might reflect BzdR-induced DNA bends leading to the loop formation.

## 4. Conclusions

In this work, we showed that the genes responsible for the anaerobic degradation of benzoate in all aromatic-degrader *Azoarcus*/*Aromatoleum* strains whose genome sequences are currently available are organized in a well-conserved *bzd* cluster. Thus, a *bzdR* regulatory gene and a *P_N_* promoter could be identified upstream of the catabolic *bzd* operon in all *bzd* clusters analyzed. Interestingly, all *P_N_* promoters show a conserved architecture that includes three BzdR operator regions (OR1–3). Whereas OR1 is sufficient for the BzdR-mediated repression of the *P_N_* promoter, the presence of OR2 and OR3 is required for the de-repression promoted by the benzoyl-CoA inducer molecule. Our results suggest that the dimeric protein BzdR binds to the *P_N_* promoter in the form of four dimers; two of the dimers are initially bound to OR1, and the complex formed may favor the interaction of two additional dimers at OR2 and OR3. The BzdR/*P_N_* complex formed induces a DNA loop that wraps around the BzdR dimers and generates a superstructure that was observed by AFM. This three-dimensional (3D) configuration may keep the −10/−35 sequences of *P_N_* inaccessible to the RNAP for achieving an efficient repression and, at the same time, could facilitate the *P_N_* de-repression when benzoyl-CoA is generated. However, the elucidation of the molecular mechanism underlying the participation of OR2 and OR3 in the de-repression of *P_N_* requires further research. In this sense, it should be taken into account that regulatory proteins other than BzdR, e.g., AcpR and AccR, are also involved in the control of the *P_N_* promoter [24,25] and, hence, more complex structure–function relationships at the *P_N_* promoter should be expected. In any case, the work presented here strongly suggests the existence of a common BzdR-dependent mechanism to control the expression of the *bzd* genes in *Azoarcus*/*Aromatoleum* strains.

## Figures and Tables

**Figure 1 genes-10-00489-f001:**
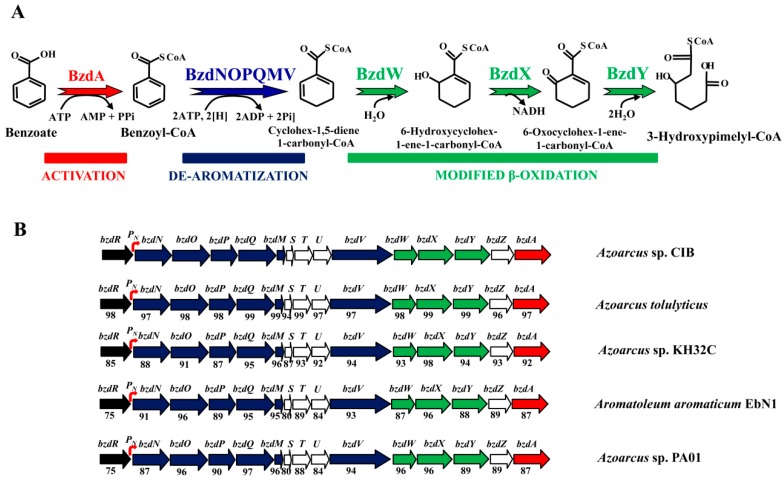
Scheme of anaerobic degradation of benzoate and gene organization of the *bzd* cluster in different *Azoarcus* and *Aromatoleum* strains. (**A**) Scheme of the anaerobic degradation pathway of benzoate into benzoyl-CoA (red), the de-aromatization of benzoyl-CoA (blue), and the modified β-oxidation that produces 3-hydroxypimelyl-CoA. Enzyme abbreviations: BzdA, benzoate-CoA ligase; BzdNOPQMV, benzoyl-CoA reductase, ferredoxin, and putative reduced nicotinamide adenine dinucleotide phosphate (NADPH)::ferredoxin oxidoreductase; BzdW, cyclohex-1,5-diene-1-carbonyl-CoA hydratase; BzdX, 6-hydroxycyclohex-1-ene-1-carbonyl-CoA dehydrogenase; BzdY, 6-oxocyclohexene-1-ene-carbonyl-CoA hydrolase. (**B**) Scheme of the *bzd* cluster in different *Azoarcus*/*Aromatoleum* strains. Genes are indicated in the same color code as the corresponding enzymes in panel (**A**), i.e., genes encoding the activation, de-aromatization, and modified β-oxidation are indicated in red, blue, and green color, respectively. The *bzdR* regulatory gene is shown in black, and the catabolic *P_N_* promoter is shown in red as a curved arrow. Genes of unknown function are colored in white. Below each gene, the percentage of the amino-acid sequence identity to the corresponding *Azoarcus* sp. CIB ortholog is indicated. The accession numbers of the corresponding genome sequences are as follows: *Azoarcus* sp. CIB (CP011072), *Azoarcus* sp. KH32C (AP012304), *Aromatoleum aromaticum* EbN1 (CR555306), *Azoarcus tolulyticus* strain ATCC51758 (NZ_FTMD00000000), and *Azoarcus* sp. PA01 (NZ_LARU00000000).

**Figure 2 genes-10-00489-f002:**
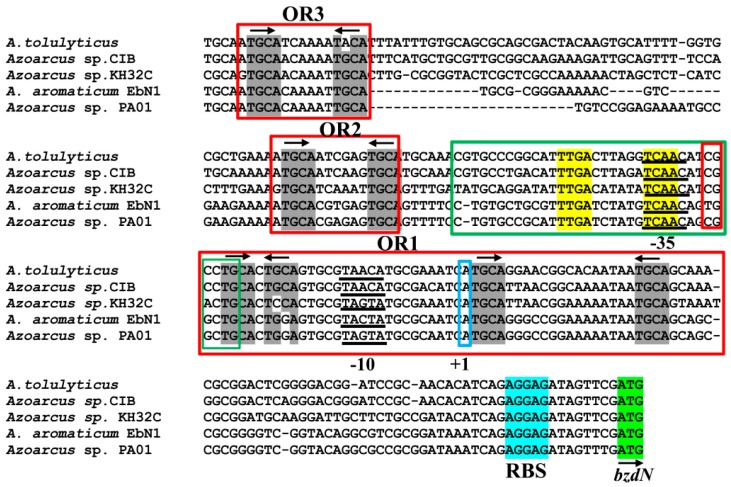
Sequence comparison analysis between the *Azoarcus* sp. CIB *P_N_* promoter [20] and the *P_N_* promoters from other *Azoarcus*/*Aromatoleum* strains. The sequence of the *P_N_* promoter from strain CIB spanning from position −174 to +79, with respect to the transcription start site (+1, indicated) is shown and compared with that of *P_N_* promoters from other *Azoarcus* strains. The σ^70^-RNA polymerase (RNAP) recognition sequences −10 and −35 are underlined, the ribosome-binding sequence (RBS) is marked in cyan, and the ATG initiation codon of the *bzdN* gene is shown in green. The operator regions I, II, and III (OR1, OR2, and OR3), involved in the interaction with the BzdR protein, are boxed in red. The arrows indicate the palindromic TGCA sequences (in gray) present in each of the operator regions. The green box shows the AcpR-binding region (palindromic sequences are indicated in yellow).

**Figure 3 genes-10-00489-f003:**
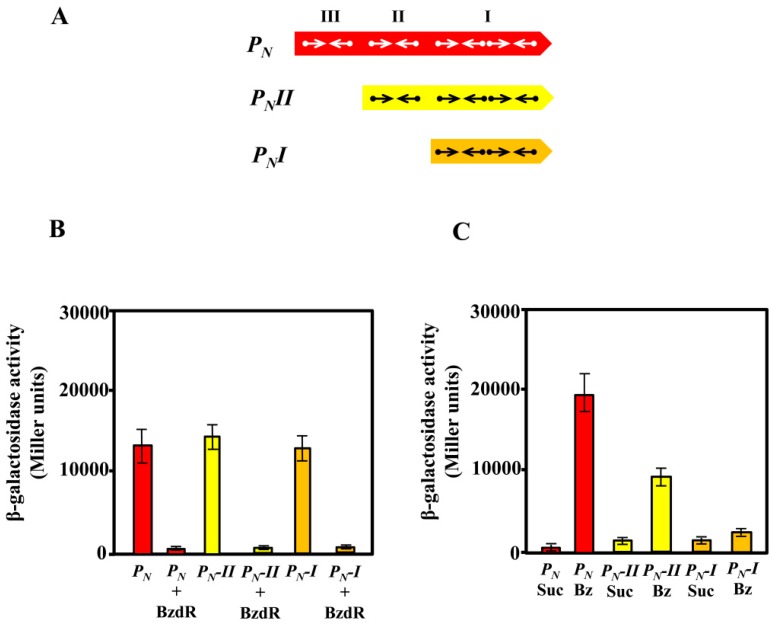
Activity of the *P_N_*, *P_N_I*, and *P_N_II* promoters in *Escherichia coli* and *Azoarcus* sp. CIB cells. (**A**) Schematic representation of the *P_N_*, *P_N_II*, and *P_N_I* promoters. Arrows represent the inverted repeat TGCA sequences for BzdR binding at operator regions I, II, and III. (**B**) The activity of the promoters was analyzed in *E. coli* MC4100 cells harboring the control plasmid pCK01 or plasmid pCK01-BzdR that expresses the *bzdR* gene (BzdR), and the plasmids pSJ3-P_N_, pSJ3-P_N_I, or pSJ3-P_N_II that harbor the *P_N_::lacZ* (*P_N_*), *P_N_I::lacZ* (*P_N_-I*), or *P_N_II::lacZ* (*P_N_-II*) translational fusions, respectively. Growth was performed in lysogeny broth (LB) in anaerobic conditions for 16 h as described in Section 2.6. The β-galactosidase activity was measured as detailed in Section 2.6 [34]. Graphed values are the averages from three independent experiments ± SD (error bars). (**C**) Activity of the *P_N_*, *P_N_II*, and *P_N_I* promoters in *Azoarcus* sp. CIB cells. *Azoarcus* sp. CIB cells harboring plasmids pBBR5-P_N_, pBBR5-P_N_I, or pBBR5-P_N_II, which express the *P_N_::lacZ* (*P_N_*), *P_N_I::lacZ* (*P_N_-I*), or *P_N_II::lacZ* (*P_N_-II*) translational fusions, respectively, were anaerobically grown for 72 h in MC medium supplemented with 0.2% succinate (Suc) or 3 mM benzoate (Bz), and the β-galactosidase activity was measured as detailed in Section 2.6. Graphed values are the averages from three independent experiments ± SD (error bars).

**Figure 4 genes-10-00489-f004:**
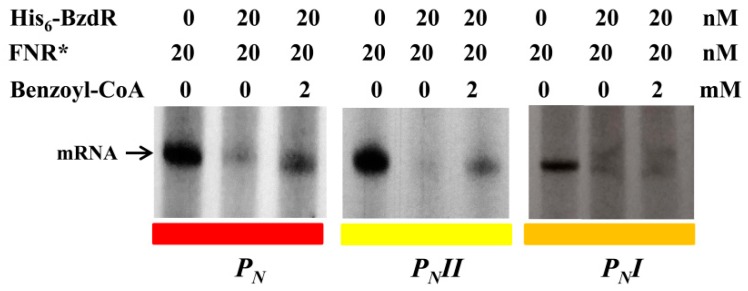
In vitro activity of the *P_N_*, *P_N_II*, and *P_N_I* promoters. Multiple-round transcription reactions were carried out as detailed in Section 2.7 by using pJCD-P_N_, pJCD-P_N_I, and pJCD-P_N_II plasmids harboring *P_N_*, *P_N_I*, and *P_N_II* promoter templates, respectively, which produced the corresponding messenger RNA (mRNA). All the in vitro transcription reactions were performed with 100 nM *E. coli* σ^70^-RNAP holoenzyme. Purified FNR* was used at 20 nM. When required, His_6_-BzdR protein was used at 20 nM, and benzoyl-CoA was added at 2 mM.

**Figure 5 genes-10-00489-f005:**
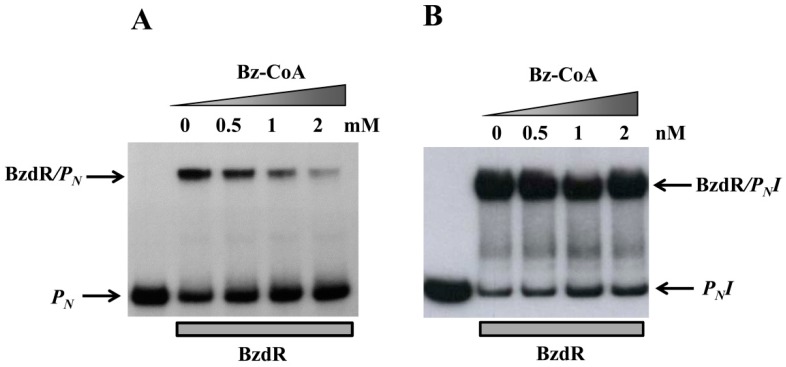
Gel retardation analysis of BzdR binding to the *P_N_* (**A**) or the *P_N_I* promoter (**B**). Gel retardation analysis was performed as described in Section 2.8 by using 10 nM purified His_6_-BzdR in the presence of increasing concentrations (from 0 to 2 mM) of benzoyl-CoA (Bz-CoA). The free *P_N_* probe and the BzdR/*P_N_* complex (**A**) or *P_N_I* probe and the BzdR/*P_N_I* complex (**B**) are indicated by arrows.

**Figure 6 genes-10-00489-f006:**
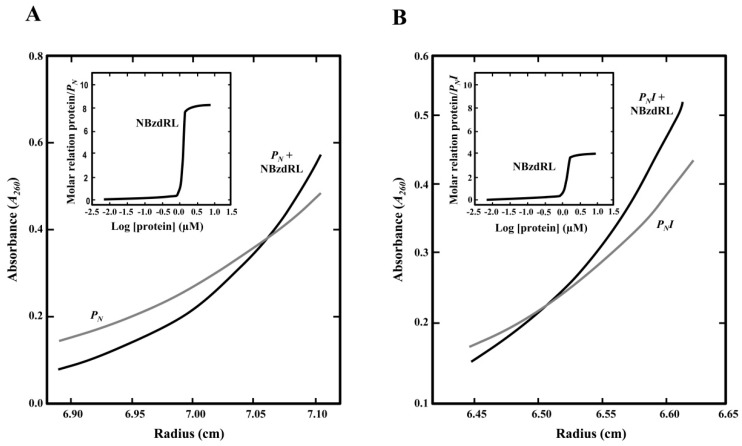
Sedimentation equilibrium analysis to study the interaction of purified NBzdRL protein with *P_N_* and *P_N_I* DNA fragments. (**A**) Distribution of the gradient in sedimentation equilibrium of the *P_N_* fragment (*P_N_*, gray line) or *P_N_* fragment with His_6_-NBzdRL protein (black line). The inset graph shows the protein/DNA molar ratio as the concentration of His_6_-NBzdRL increases. (**B**) Distribution of the gradient in sedimentation equilibrium of the *P_N_I* fragment (gray line) and *P_N_I* + His_6_-NBzdRL complex (black line). The inset graph shows the protein/DNA molar ratio as the concentration of NBzdRL increases.

**Figure 7 genes-10-00489-f007:**
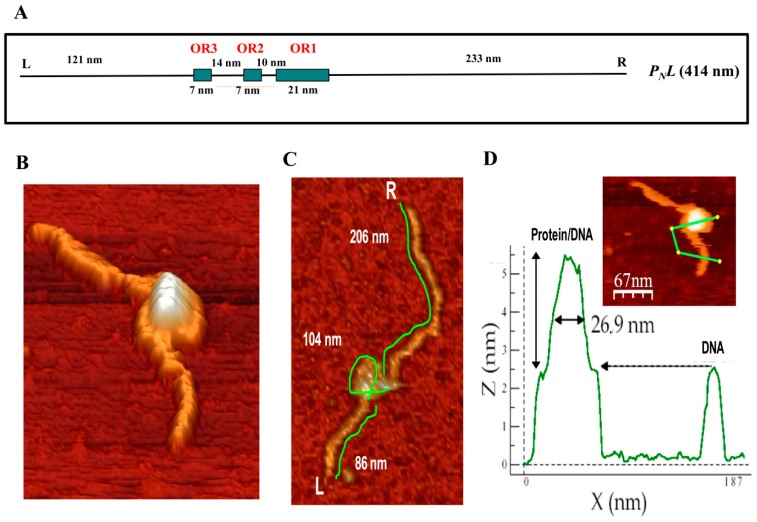
Analysis of the binding of BzdR to the *P_N_* promoter by atomic force microscopy (AFM). (**A**) Scheme of the *P_N_L* DNA fragment used as a template. The DNA fragment extends from position −505 to +720 with respect to the transcription start site of the *P_N_* promoter. The operator regions recognized by BzdR are represented by green boxes with their corresponding numbers (OR1, OR2, and OR3). The distances separating each of the indicated regions, the extension of each operator region in nm, and the left (L) and right (R) ends of the *P_N_L* fragment are also indicated. (**B**) Image of a superstructure of the BzdR/*P_N_L* complex. (**C**) Analysis of distances in a BzdR/*P_N_L* superstructure (one image, *n* = 16). The distances of both arms and the loop of the superstructure are detailed. L and R indicate the left and right ends of the *P_N_L* DNA fragment, respectively. (**D**) Image of a BzdR/*P_N_L* superstructure and its corresponding height profile. The green line of the image corresponds to the profile outlined in the lower graph. In the graph, the length (in nm) of the line is represented on the *x*-axis and the height on the *y*-axis. The first peak corresponds to the BzdR/*P_N_L* superstructure and the second peak corresponds to the naked DNA.

**Table 1 genes-10-00489-t001:** Bacterial strains and plasmids used in this study.

Strain or Plasmid	Relevant Genotype and Characteristic(s)	Reference
***Escherichia coli* strain**		
DH5α	*endA1 hsdR17 supE44 thi-1 recA1 gyrA*(Nal^r^) *relA1* Δ(*argF-lac*) *U169 depR Φ80dlacd*(*lacZ*) *M15*	[26]
S17-1λ*pir*	Tp^r^ Sm^r^ *recA thi hsdRM^+^* RP4::2-Tc::Mu::Km λ*pir* phage lysogen	[27]
M15	Strain for regulated high-level expression with pQE vector	Qiagen *
MC4100	F^-^, *araD319*, *Δ(argF-lac)U169 rpsL150* (Sm^r^) *relA1 flbB5301 deoC1 ptsF25 rbs*	[28]
***Azoarcus* strain**		
CIB	Wild-type strain	[23]
**Plasmids**		
pQE32	Ap^r^, *ori*ColE1 T*5* promoter *lac* operator, *λt_0_*/*E. coli rrnB T1* terminators, N-terminal His_6_	Qiagen *
pQE32-His_6_BzdR	Ap^r^, pQE32 derivative harboring the His_6_-*bzdR* gene	[20]
pQE60-His_6_Fnr*	Ap^r^, pQE60 derivative harboring the His_6_- FNR* gene under the control of *T5* promoter *lac* operator	[29]
pQE32-His_6_NBzdRL	Ap^r^, pQE32 derivative harboring the His_6_- NBzdRL fragment	[30]
pREP4	Km^r^, plasmid that expresses the *lac I* repressor	Qiagen *
pECOR7	Ap^r^, pUC19 harboring a 7.1-kb *Eco*RI DNA fragment containing the *bzdRNO* genes	[23]
pBBR1MCS-5	Gm^r^, oripBBR1MCS Mob^+^ *lacZα*, broad-host- range cloning and expression vector	[31]
pBBR5P_N_	Gm^r^, pBBR1MCS-5 derivative harboring a *P_N_::lacZ* translational fusion	[20]
pBBR5P_N_I	Gm^r^, pBBR1MCS-5 derivative harboring a *P_N_I::lacZ* translational fusion	This work
pBBR5P_N_II	Gm^r^, pBBR1MCS-5 derivative harboring a *P_N_II::lacZ* translational fusion	This work
pJCD01	Ap^r^, *ori*ColE1, polylinker of pUC19 flanked by *rpoC* and *rrnBT1T2* terminators	[32]
pJCD-P_N_	Ap^r^, pJCD01 derivative harboring a 585-bp *Eco*RI fragment that includes the *P_N_* promoter	[24]
pJCD-P_N_I	Ap^r^, pJCD01 derivative harboring a 139-bp *Eco*RI fragment that includes the *P_N_I* promoter	This work
pJCD-P_N_II	Ap^r^, pJCD01 derivative harboring a 209-bp *Eco*RI fragment that includes the *P_N_II* promoter	This work
pSJ3-P_N_	Ap^r^, pSJ3 derivative harboring a *P_N_::lacZ* translational fusion	[20]
pSJ3-P_N_I	Ap^r^, pSJ3 derivative harboring a *P_N_I::lacZ* translational fusion	This work
pSJ3-P_N_II	Ap^r^, pSJ3 derivative harboring a *P_N_II::lacZ* translational fusion	This work
pCK01	Cm^r^, *ori*pSC101, low copy number cloning vector, polylinker flanked by *Not*I sites	[33]
pCK01-BzdR	Cm^r^, pCK01 derivative harboring a DNA fragment containing the *bzdR* gene	[20]

Ap^r^: ampicillin resistant; Km^r^: kanamycin resistant; Cm^r^: chloramphenicol resistant; Gm^r^: gentamicin resistant. *: Qiagen, Hilden, Germany.

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
