# Peer review of "Further Insights into the Architecture of the PN Promoter That Controls the Expression of the bzd Genes in Azoarcus"

_genes, 2019, doi:10.3390/genes10070489_

Round 1
Reviewer 1 Report
The 'Comments and Suggestions for Authors' are summarized in the uploaded pdf-file.

Author Response
We would like to thanks the labour of the reviewers to importantly improve the submitted version with their criticisms. The new version includes modification in 6 figures, new experiments, recalculation of some data and more literature-based information. Find enclosed the answer point-by-point to all the questions raised by the reviewers.
Yours sincerely,
Manuel Carmona
REVIEWER 1
The manuscript entitled ‘New insights into the BzdR-mediated transcriptional control of the bzd genes for the anaerobic catabolism of benzoate in Azoarcus’ by Gonzalo Durante-Rodríguez and coworkers revealed that the promoter region located upstream of the bzdN gene is conserved among benzoate-degrading Azoarcus strains. To study the function of three known operator regions of this promoter (PN), two truncated promoters, PNI and PNII, were designed and characterized. Furthermore, the complex of PN and the regulator protein BzdR was visualized by atomic force microscopy for the first time.
General comments
Please specify the title of the manuscript. The current title is almost identical to the title of a previously published study of your group (Durante-Rodríguez G et al., 2008; Microbiology 154, 306-316; ‘New insights into the BzdR-mediated transcriptional regulation of the anaerobic catabolism of benzoate in Azoarcus sp. CIB).
A: According to the reviewer´s suggestion, we have changed the title of the manuscript. The new title is: “Further insights into the architecture of the PN promoter that controls the expression of the bzd genes in Azoarcus”.
Please clearly state in the abstract, which of the provided information is new or already known from previous studies. For me, it sounds that the PN promoter (line 16) was identified here for the first time (what is wrong). Further, the new data about the function of OR1 to OR3 (line 19-21) obtained by truncation of PN seem to be known before.
A: We have rewritten the abstract to clarify the information that was previously known (lines 15-20).
In the Material and Methods section, some companies are listed. Please also include city and country information. In general, the Material and Methods section should be checked for correctness (see also specific comments).
A: Ok, the information regarding the city and country of the companies listed in the manuscript has been added.
Specific comments
Line 21/22: ‘…in the form of four dimers, two of them binding to OR1.’
A: OK; done (line 22).
Line 33: Fungi (white rot fungi) are also very important degraders of aromatic compounds, especially of complex structures. Bacteria are more specialized on simple aromatics.
A: OK, fungi are also mentioned (line 34).
Line 34-36: Please provide as reference for this statement review articles about the aerobic and anaerobic pathways studied so far.
A: Ok, three new reviews dealing with the degradation of aromatic compounds in aerobic conditions have been added (line 36), and also a new review dealing with anaerobic metabolism of aromatic compounds was included to emphasize the argument used in this paragraph (line 39).
Line 42: Besides benzoyl-CoA also derivatives of this compound can be formed as central intermediate.
A: Yes the reviewer is right. In fact, this concept can be inferred from the sentence: “.... a wide variety of aromatic compounds converges into a few central pathways”. However, in this manuscript we wanted to focus on those aromatic compounds that converge into benzoyl-CoA, as stated in the sentence “most monocyclic aromatic compounds are channelled and activated...to benzoyl-CoA” but it is clear that some others are converted to intermediaries other than benzoyl-CoA (lines 40-44).
Line 42: [2, 5-6]
A: OK, the parenthesis has been deleted (line 43).
Line 42/43: change to ‘…as a model compound to study the main pathway for anaerobic degradation of aromatic compounds, …’
OK, done (lines 43-45).
Line 47: include comma before ‘which’
A: Ok, done (line 49).
Line 44-55: In this chapter, the anaerobic degradation of benzoate is briefly explained. Please include that two different types of benzoyl CoA reductases are known. Since Azoarcus harbors type I, this type (reaction and corresponding genes) was visualized in Figure 1 (please indicate in figure legend). Do all the transcriptional regulators previously studied (line 52-55) belong to type I benzoyl CoA reductases?
A: (1) We have not discussed on the type of enzymes (benzyl-CoA reductases included) involved in the anaerobic degradation because this work is not dealing with metabolism. Instead, we focused on the genetic organization of the bzd clusters from Azoarcus-like bacteria and their regulation. (2) The transcriptional regulators BzdR and BadM control clusters that encode type I-BCRs whereas BamVW and BgeR regulate clusters that encode type II-BCRs. Since these four regulators belong to different regulatory families, it seems that there is not a clear correlation between a particular type of transcriptional regulator and a specific type of BCR.
Line 52: cite reference 5 and 7 (in place of 4 and 7)
A: Ok, done (line 54).
Line 57: remove comma after benzoate
A: Ok, done (line 51).
Line 59, line 65: β-oxidation
A: Ok, done (line 62).
Line 61: benzoyl-CoA reductase
A: Ok, done (line 63).
Line 63: Please check the name of the enzyme BzdY. According to the figure it is a 6-ketocyclohex-1-ene-1-carbonyl-CoA hydrolase. Commonly the enzyme is termed 6-oxocyclohex-1-ene-1-carbonyl-CoA hydrolase.
A: Ok, done (line 65).
Line 67: In Figure 1B, the promoter as well as the bzdA gene is shown in red. Possibly, use the wording: ‘… promoter is shown in red as curved arrow.’
A: Ok, done (line 70).
Line 68/69: Please rephrase: ‘Below each gene the percentage of the amino acid sequence identity to the corresponding Azoarcus sp. CIB ortholog is indicated.’
A: Ok, done (lines 70-71).
Line 74: ‘represent’ in place of ‘constitute’?
A: Ok, done (line 77).
Line 82-84: I had to read the sentence several times until I got the information. Simplify to: ‘Besides BzdR two additional regulatory proteins … :’
A: Ok, the sentence was changed as indicated (line 87).
Line 86: ‘Further’ in place of ‘On the other hand’
A: Ok, done (line 90).
Line 89: ‘the’ in place of ‘our’
A: Ok, done (line 93).
Line 89: What do you mean with ‘transcriptional organization’?
A: It makes reference to the organization of the cluster. The word “transcriptional” has been removed to make it clear (lines 93-94).
Line 91: The three operator regions of the PN promoter were previously identified in Azoarcus sp. CIB. This information has to be provided in the introduction section (before this paragraph) together with an appropriate reference.
A: Ok, done (lines 85-86).
Line 99: What do you mean with ‘the corresponding necessary nutritional supplements’? Please specify.
A: The nutritional supplements are casamino acids 0.1%. These specifications have been added in the new version of the manuscript (line 102).
Table 1: In the present version of the manuscript, some text/numbers are shifted. Please indicate as footnote the meaning of Apr and Kmr.
A: Ok, done (Table 1).
Line 121: singular, ‘strain’
A: Ok, done (Table 1).
Line 127: Apr
A: Ok, done (Table 1).
Line 170: ‘… were carried out according to published methods [24].’
A: Ok, done (line 164).
Line 172/173: Check wording. The sequence of inserts/DNA fragments was confirmed.
A. Ok, done (line 167-168).
Line 174: What do you mean with ‘RbCl method’?
A: The sentence was completed as: “Transformation of E. coli was carried out by using competent cells prepared by the RbCl method [34] (lines 171-172).
Line 177: [21].
A: Ok, done (line 172).
Line 181/182: Please specify (e.g. sequences of proteins encoded in the bzd operon).
A: Ok, done (line 179).
Line 186-188: Genes are expressed. Proteins are produced. You combined ‘express’ and ‘protein’. Please correct. Shortly explain the characteristics of the NBzdRL protein (N-terminal domain + linker).
A: Ok, done (line 185).
Line 193: Check wording. ‘To perform experiments of and …’?
A: Ok, the word “and” has been deleted (line 192).
Line 195: ‘DNA probe’ is misleading. This term is usually used for labeled DNA.
A: Ok, “DNA probe” has been replaced by “DNA fragment” along the manuscript when the fragment described was not labelled.
Line 200: ‘were conducted in a buffer containing …’
A: Ok, done (line 199)
Line 201: … [19]. The presence of DNA was checked by measuring …’
A: Ok, done (line 200).
Line 202: Please specify: ‘the different species obtained’?
A: Ok, the sentence was rewritten (lines 200-201).
Line 207: For the truncated promoters, refer to Figure 3A.
A: Ok, done (line 206).
Line 214: ‘in-frame’ in place of ‘in phase’
A: Ok, done (line 217).
Line 217: Check wording. ‘rendering’?
A: Ok, “generating” was used instead of “rendering” (line 220).
Line 220: Was E. coli cultivated in the presence or absence of oxygen?
A: E. coli was grown in anoxic conditions. It has been mentioned in the new version (line 223).
Line 223/224: Please extend the reference 32 by the page numbers of the book chapter(s), which is (are) important for this experiment (in the reference list). How the cells were permeabilized? Was the method similar for E. coli and Azoarcus?
A: Ok, the page numbers were included in the reference. The cells were permeabilized by using SDS 0.1% and chloroform (indicated now in the manuscript) as detailed in reference 34 (lines 226-228). The method was the same for E. coli and Azoarcus.
Line 228: ‘DNA fragment’ (singular)
A: Ok, done (line 231)
Line 229: Is there a special reason, why the template for the amplification of the promotor region was on the one hand the pECOR7 plasmid (Line 212) and on the other hand genomic DNA (Line 229)?
A: There is not a special reason; we used either pECOR7 or Azoarcus sp. CIB genomic DNA as templates for PCR amplification.
Line 229-235: Check sentence. First, the DNA fragments have to be amplified, then restriction has to be performed and finally the fragments will be cloned into the plasmid.
A: Ok, the sentence has been rewritten (line 237).
Line 232: The oligonucleotide sequence of 3IVTPN is missing.
A: Ok, the sequence has been added (line 235).
Line 226: For in vitro transcription analysis, please also cite Claverie-Martin, F. & Magasanik, B. (1992). https://doi.org/10.1016/0022-2836(92)90516-M.
A: Ok, done (line 229).
Line 239/240: Check the concentrations for ATP, CTP, GTP and UTP. I guess ‘μM’ will be correct. For UTP, the stock is probably 3,000 Ci/mmol. Please also check.
A: The reviewer is right. The NTPs concentration was in the μM range and the activity of the labeled UTP stock was 3000 Ci/mmol. It has been corrected in the text (lines 244-245).
Line 242: Which carrier tRNA was used?
A: We use yeast tRNA from ThermoFisher Scientific. The information was incorporated in the manuscript (lines 247).
Line 242: ‘separated’ in place of ‘electrophoresed’
A: Ok, done (line 248).
Line 245: Which method was used for separation and visualization of the bands of the gel retardation assay? (the same as described directly above?)
A: No, a sentence explaining the method has been added: “The samples were fractionated by electrophoresis in 5% polyacrylamide gels buffered with 0.5 X TBE (45 mM Tris borate, 1 mM EDTA). The gels were dried onto Whatman 3MM paper and exposed to Hyperfilm MP (Amersham Biosciences, Little Chafont, UK).” (lines 258-261).
Line 246: PNI
A: Ok, done (line 252).
Line 249: Which proteins were used? State the concentrations of the proteins or revise sentence (‘… used in each assay as described before.’?)
A: OK, the sentence has been rewritten (line 256).
Line 250: ‘…20 mM Tris HCl pH 7.5 …’ (check throughout, no comma between Tris and ph value)
A: Ok, done (line 256).
Line 254: ‘DNA probe’ is misleading. This term is usually used for labeled DNA.
A: OK, it was substituted for “DNA fragment” along the text.
Line 256: ‘PCR-amplified’ in place of ‘PCR-obtained’
A: Ok, done (line 259).
Line 258: Remove ‘on the other hand’
A: Ok, done (line 264).
Line 259: remove comma before ‘using’
A: Ok, done (line 266).
Line 261: What means ‘adequate’?
A: The sentence has been rewritten to avoid misinterpretations: “The protein and DNA concentrations were optimized to ensure that the unbound protein ... did not disturb the image” (line 275-278).
Line 263: ‘probe’ is misleading. See above.
A: Ok, “fragment” was used instead of “probe” (line 264).
Line 263-267: Check order: Mixtures have to be prepared, dilution in buffer containing MgCl2 and NiCl2, application on surface, washing, … .
A: Ok, done (lines 270-275).
Line 283: β-oxidation
A: Ok, done (line 293).
Line 286: What do you mean with ‘conserved transcriptional organization’? Among Azoarcus, certain sequence motifs are conserved in this promoter region.
A: Ok, the sentence was changed by “this transcriptional organization found in Azoarcus was not observed in closely related aromatic…” (lines 294-296).
Line 287: ‘… in closely related aromatic compound degrading denitrifying bacteria such as those of the Thauera genus.’
OK, done (line 298).
Line 286/287: Please cite a reference about the “transcriptional organization” in Thauera species. Do Thauera species harbor a bzdR homolog? What ‘was not observed’ in Thauera species?
A: As far as we know, the transcriptional regulation of the cluster bcr/bzd from Thauera has not been described in any publication. We perform a search in the genome sequence of Thauera aromatica and other sequenced Thauera strains of bzdR and PN orthologs, and we found that both sequences were not present in any Thauera genome. We have highlighted in the new version (line 296-298) that a sequence similar to PN was not observed in Thauera species.
Line 288-294: It is not obvious that this information is already known from previous studies. Include words like ‘previously’ or ‘before’. Cite the corresponding references (which are already provided in the text) also in the legend of Figure 2. The new information is the conservation of this region in various Azoarcus strains.
A: Ok, done (line 299).
Line 288-291: ‘In the PN promotor of strain CIB, the transcriptional start site is located 75 nucleotides upstream of the ATP start codon of the bzdN gene.’ In the two following lines the term ‘mapped’ sounds odd to me. Please check.
A: Ok, the terms “identified” and “detected” were used instead of “mapped” (lines 299-301).
Line 294: (green box, Figure 2)
A: Ok, done (line 323-324).
Line 296: Is anything known about the interaction of AccR with the promoter region?
A: Yes, AccR binds to PN promoter as mentioned in the Introduction and referenced there [24] (lines 87-94).
Line 297: The (N6) is only true for OR2 and OR3. The given palindromic sequence is true for OR2 and OR3 with one exception.
A: Ok, it was corrected (lines 306-312).
Line 298: N6 is A-rich only in OR3.
A: Ok, it was corrected (lines 306-312).
Line 299: The palindromic sequence TGCA(C)TGCA is not conserved to 100%. The nucleotide sequence at position 7 and 8 can be GC, CC or GG.
A: Ok, it was corrected (line 311).
Line 299: TGCA(N15)TGCA
A: Ok, done (line 311).
Line 306: ‘marked in cyan’
A: Ok, done (line 320).
Line 311: In which organisms the TGCA site is favored? Is it specific for Bacteria or for all kingdoms of life? In the following lines a few examples were mentioned. Why selecting CopG? Is it related to the study? Also include the organism, in which CopG is present.
A: We have decided to eliminate the first sentence of the paragraph (“The TGCA sequence has been relatively favoured throughout evolution...”) since the work was cite in Ref. 40 dealed with eukaryotic examples (the reference has been also removed). We maintained a few bacterial examples to illustrate that BzdR is not the only prokaryotic transcriptional regulator able to recognize TGCA sequences (lines 317-326). CopG,, XylS or BenR were selected as examples. CopG is present in the genus Steptococcus.
Line 319: TGCA(N6)GGNTA
A: Ok, done (line 332).
Line 329: ‘…overlapping the RNAP -35 box …’
A: Ok, done (line 343).
Line 331: Who or what possibly blocks the interaction with RNAP? I do not understand the connection to the sentence before.
A: The sequence was substituted for: “having the requested sequence for the interaction with RNAP” (line 348).
Line 334: remove ‘we have’ after ‘and’
A: Ok, done (line 353).
Line 336: ‘and truncated PNII promoter’
A; Ok, done (line 353).
Line 340/341: Remove ‘two strains that carry the empty plasmid pCK01 or the pCK01 plasmid 6 containing the bzdR gene, respectively’ (the same information twice)
A: Ok, done (line 356).
Line 343: How this promoter can be active in E. coli when AcpR is missing? (see Line 328) Briefly explain.
A: AcpR is a protein that belongs to the FNR/CRP superfamily. We have shown previously that the FNR protein from E. coli is able to bind and activate the PN promoter (reference 24). We introduce this sentence in the text (lines 340-343), and we also introduce the reference that describes the FNR/CRP superfamily [49].
Figure 3: Use in B and C the same scale for the y-axis for direct comparison. Please check the calculation of the Miller units: previously, an activity of about 2,000 Miller Units was determined for the PN promoter (see DOI: 10.1128/JB.188.7.2343-2354.2006). Think about the order of the columns in B and C. For me, it makes more sense to order the columns according to the length of the promoter, as it was also done in Figure 3A (sorted from the full-length promotor to the shortest, or the other way around).
A: Ok, (i) B and C are now at the same scale, (ii) the differences most probably are due to the strain used. In JB2006 was used the E. coli M182 strain, whereas here was used the E. coli MC4100 strain. In a previous work where MC4100 strain was used, around 9,000 Miller Units were obtained (reference 30); (iii) the order of the columns has been changed (Figure 3).
Line 356/361: There is no detailed description available in the Material and Methods section. Please state the reference.
A: Ok, the description of the anaerobic growth of E. coli in rich medium has been added in Material and Methods (lines 101-102).
Line 364: Include one or two connecting sentences between line 347 and line 364. Starting form line 364, the experiment was done in Azoarcus. In contrast to E. coli, another “regulating” agent, in particular benzoyl-CoA, is present depending on the growth conditions. This fact is briefly mentioned in the introduction and should be resumed here. Then, it will be possible for the reader to quickly understand the term ‘de-repression’ (Line 364 and ongoing).
A: Ok, the sentences “Azoarcus sp. CIB is able to produce benzoyl-CoA when benzoate is present in the culture medium. Benzoyl-CoA acts as inducer molecule allowing the de-repression of the system” were introduced to answer the question addressed by the reviewer (lines 382-385).
Line 366: Check wording. ‘rendering’?
A: Ok, “originating” was used instead of “rendering” (line 387).
Line 377: ‘almost no de-repression’
Ok, done (line 397-398).
Line 380: The term ‘de-repression’ should be followed by ‘(induction)’, when it appears for the first time in this paragraph (Line 364).
A: Ok, done (line 381).
Line 380: ‘… to be essential for the full de-repression’
A: Ok, done (line 401).
Line 387: Why FNR* in place of AcpR was applied in the assay?
A: We have demonstrated previously (reference 22) that FNR* is able to activate the promoter PN even in the presence of oxygen. The main reason to use FNR* instead of AcpR is that the transcription in vitro experiments were performed in aerobic conditions and AcpR (and also FNR) are inactive when oxygen is present. The FNR* protein is a constitutive mutant of the FNR protein that allows to perform in vitro experiments in the bench (aerobic environment). All this information is detailed in reference 24. The reference has been added to the manuscript (line 408).
Line 388: The transcript is visible, however the size cannot be deduced from the figure because no marker is shown.
A: The reviewer is right. An internal control with a transcript of 105 nt (identical to that shown in Figure 5 of reference 24) was used. We removed this control to make the image clearer.
Line 389-391: Is it possible to quantify the difference of intensity of the band showing the de-repression for the PN and PNII promoter? It is more scientific to state a value in comparison to ‘a slight de-repression’. From the figure, I would guess it is a factor of about three to five.
A: Ok, we have detailed in the text the data from the quantification done by using the Bio-Rad Molecular Imager FX system and the ImageJ software (lines 411-412). Both methods have been included in the Material and Methods section (lines 249-250).
Line 393: ‘when both, OR2 and OR3 are missing.’ Here, ‘both’ is very important. When only OR3 is missing, the de-repression is weaker compared to the entire promoter, but still possible.
A: Ok, done (line 414).
Line 394: BzdR
A: Ok, done (line 416).
Line 396-399: ‘…but increasing concentrations of benzoyl‐CoA up to 2 mM, previously shown to release the BzdR repressor from the wild‐type PN promoter [18], were unable to induce the dissociation of the BzdR/PNI complex (Figure 5).’
A: Ok, done (lines 419-421).
Line 399/400: I cannot follow this conclusion. Finally, you are right that BzrD is still bound to the promoter. However, it is not clear, why benzoyl CoA did no function as inducer for PNI. You might speculate more on this. Is there anything known from the literature how benzoyl CoA recognizes BzrD? What is the target of benzoyl CoA? …
A: Benzoyl-CoA is recognized by the C-terminal domain of BzdR and provokes a conformational change in the protein that ends in the release of the repressor from the target promoter. The detailed information regarding this phenomenon has been explained in detail in reference 18, and we have noted this in a new sentence in the text. We are not able to find yet a reason that might explain why BzdR is not released from PNI based on the interaction between BzdR-PNI-benzoyl-CoA (line 421-413).
Figure 4: Think about the order (see Figure 3). How a control without FNR* looks like?
A: Ok, the order has been changed. The control without control gives no transcript (as shown in Figure 5 of reference 24).
Line 406: FNR* was present in all assay mixtures.
A: Ok, explained in new sentence (lines 433-434).
Line 409: Include reference [19] also after ‘cooperative’.
A: Ok, done (line 436).
Line 413: ‘… when both, OR2 and OR3 are missing …’
A: Ok, done (line 440).
Figure 5: Although it was previously shown how the entire promoter PN behaves in this analysis, the experiment should have been performed in parallel as a positive control. Why PNII was not tested? What will happen when the concentration of benzoyl CoA is higher than 2 mM? It appears that there was a slight increase in unbound PNI promotor with increasing concentrations of benzoyl-CoA. What is the intracellular concentration of benzoyl CoA?
A: (i) We have included a new gel retardation experiment with the control of the entire PN promoter in the panel A of figure 5, (ii) we did not test PNII because we wanted to focus on PNI, that was the PN-derivative that showed the most clear non-depression effect (as shown in the in vitro transcription experiments), (iii) since benzoyl-CoA is a very expensive reactive and we have not observed clear differences in the gel retardation assays between 0.5, 1 and 2 mM benzoyl-CoA (which show a clear effect inhibiting binding of BzdR to the wild-type PN promoter), we decided not to use higher concentrations of this compound, (iv) as far as we know there is no report about the intracellular concentration of benzoyl-CoA. However is hard to believe concentrations in the range of mM since the concentration of CoA is in the microM range.
Line 420/421: Why only the N-terminal domain and not full length BzdR was applied?
A: We have observed that when BzdR was used in experiments of sedimentation equilibrium experiments at concentration higher than 2 microM, the protein aggregates. Since we liked to perform experiments at 10 microM protein, and NBzdRL is totally soluble at this concentration, we decided to use NBzdRL instead of BzdR for the ultracentrifugation experiments.
Line 421/422: two times PN promoter. In which experiment it was shown that NBzdRL binds to PNI?
A: In the reference [30], that has been included in the text (line 451). The duplication of PN promoter has been deleted (line 454).
Line 423: ‘single species’?
A: It emphasizes that 100% are dimers. To avoid redundancies, we have eliminated the expression “single species” (line 455).
Line 424: Which study do you mean with ‘here’?
A: The results presented in Fig. 6A, as mentioned now in the text (line 456-459).
Line 427: NBzdRL
A: Ok, done (line 457).
Line 429: Which ‘previous assumption’?
A: The conclusions obtained from the work done at reference [21]. That reference has been added at the end of the sentence (line 459).
Line 430: From my point of view, it seems conclusive that the binding of each BzdR dimer to the promotor requires a palindromic TCGA recognition site. Your experiments further highlighted this assumption. I do not see the need to introduce the hypothesis of an excess of BzdR might bind to OR1, when both, OR2 and OR3 are absent. This hypothesis is not very intuitive and causes confusion (at least, it did with me). (see also lines 412-414)
A: Ok, to avoid confusions we delete the sentence (line 459).
Figure 6: I am not familiar with this method. How did you obtained the data in the inset from the main graph? How the data of the main graph were collected? Please explain.
A: The technical details are explained in reference 21, that is included in Material and Methods (lines 202-202).
Change title: e.g. ‘Sedimentation equilibrium analysis to study the interaction of purified NBzdRL protein with the PN and PNI DNA fragments’
A: Ok, done (line 462).
Line 435: What ‘PNU’ stands for?
A: PNU was the name of the PN DNA fragment that we used for the sedimentation equilibrium studies. We used PN to avoid misinterpretations (Fig. 6).
Line 447: ‘probe’ misleading, see above
A: Ok, done (line 464 and along the figure 6 legend).
Line 448: Is the given ratio true for the number of molecules (BzdR molecules to DNA molecules)?
A: Yes, it is.
Line 451 ‘… (Figure 7C, shown for one of the 16 samples).’
A: Ok, done (lines 480-481).
Line 451/452: ‘… are slightly lower than those theoretically expected, i.e. 121 and 233 nm, respectively (Figure 7A). The …’
A: Ok, done (lines 481-483).
Line 453: How the ‘18%’ reduction of the length was calculated? I failed in getting this value. Also state a mean value plus standard deviation (as done before for the nms).
A. It was calculated comparing the length of the emerged DNA at both sides of the compact structure (108.9 ± 20 nm (L) and 199.9 ± 44 nm (R) with the length (L (121 nm) and R (233 nm) of the naked PNL fragment. We have re-calculated the reduction in length and the reviewer is right, the value obtained is 14% ± 4%. This value has been introduced in the new version (line 482).
Figure 7: Did you also analyzed the DNA template (without BzdR) by this method. If yes, was the size as expected?
A: Yes, naked PNL DNA fragment were observed by AFM and their length was measured with an estimated size of 400 nm ± 13 nm (Gutiérrez-Arroyo, P. Ph D. Thesis).
Line 457: ‘probe’ misleading, see above
A: Ok, done (changed along the text).
Line 461: ‘… superstructure (one example, n =16). The distances …’
A: Ok, done (line 492).
Line 462: Since DNA was used in this assay, I would recommend 5’ and 3’ prime end and not left and right.
A: Since we are using double strand DNA we think that the annotation 5’ and 3’ is not accurate. We have introduced the L/R annotation in the Figure 7A to make it clear.
Line 470: ‘estimated to be 398.6 …’
A: Ok, done (line 501).
Line 488: ‘… could by identified …’
A: Ok, “could be” used instead of “can be” (line 520).
Line 498: Is there any information available about the mechanism of activation by benzoyl CoA? If so, please include in the Results and Discussion section.
A: Our results suggest that benzoyl-CoA induce conformational changes in BzdR leading to the release of the repressor from the promoter (reference 21). Sentence added at lines 384-386.
Reviewer 2 Report
The manuscript by Durante-Rodríguez et al. presents an extensive analysis of the DNA binding mode of the repressor of anaerobic benzoate metabolism, BzdR. The authors show clearly that an extended operator region containg the boxes OR1-3 is required for derepression by benzoyl-CoA and that BzdR dimers are binding at every box, forming a large DNA-bound complex. There are only minor points of criticism, as listed below:
l. 51: since most of the aromatic-degrading Azoarcus species have recently been formally renamed Aromatoleum, this genus should be listed here as well.
Fig. 1: the newly assigned genus names should be used here
l 199-201 not a full sentence and dot missing
l 281: bzdR as written here represents the gene, not the enzyme generating benzoyl-CoA
i 283: greek beta, and again, the genes are mis-taken for the protein functions
l 287 (and other places): reverse to "genus Thauera" for proper English
l 288: replace "at 75 nucleotides" by "minus 75 ..." or "75 nt 5' to..."
l 291 in or to, but not into
Fig. 2: the rbs indicated appears too long, Usually, only the longest stretch of complementary sequence to the 3'-end of 16S rRNA is taken into account (AGGAG in this case)
l 332 and 412: replace thought by expected
l 369: "Azoarcus cells ... led to ... activity": poor English results in logical mess-up
l 386: using the ...plasmids... as DNA templates
l. 390 slight
l 393: at promoter PNI
Fig. 5: this is my only substantial criticism: it is not sufficient to just show the behaviour of the one construct where nothing happens. This kind of experiment has to be acompanied by a control with the full promoter where the addition of B-CoA releases the binding. The authors can not refer to a previous paper here, but have to show that the constructs behave differently under identical experimental conditions (possible day-by-day variations may be caused e.g. by different degradation grades of protein or different hydrolysis ratios of B-CoA).
l 423: capital S for Svedberg constant
l 430: do you mean disprove rather than discard?
l 493: the dimeric protein BzdR
Author Response
Find enclosed the new version of the manuscript entitled “Further insights into the architecture of the PN promoter that controls the expression of the bzd genes in Azoarcus” (reference ID#502135). We would like to thanks the labour of the reviewers to importantly improve the submitted version with their criticisms. The new version includes modification in 6 figures, new experiments, recalculation of some data and more literature-based information. Find enclosed the answer point-by-point to all the questions raised by the reviewers.
Yours sincerely,
Manuel Carmona
REVIEWER 2
Comments and Suggestions for Authors
The manuscript by Durante-Rodríguez et al. presents an extensive analysis of the DNA binding mode of the repressor of anaerobic benzoate metabolism, BzdR. The authors show clearly that an extended operator region containg the boxes OR1-3 is required for derepression by benzoyl-CoA and that BzdR dimers are binding at every box, forming a large DNA-bound complex. There are only minor points of criticism, as listed below:
l. 51: since most of the aromatic-degrading Azoarcus species have recently been formally renamed Aromatoleum, this genus should be listed here as well.
A: Ok, done (line 53).
Fig. 1: the newly assigned genus names should be used here
A: Ok, done at (lines 60 and 66).
l 199-201 not a full sentence and dot missing
A: Ok, done (line 206-209).
l 281: bzdR as written here represents the gene, not the enzyme generating benzoyl-CoA
A: Right, bzdA is the gene that code for the enzyme that generates benzoyl-CoA. It has been detailed in the new version (lines 291-294).
i 283: greek beta, and again, the genes are mis-taken for the protein functions
A: Ok, done (Lines 291-296).
l 287 (and other places): reverse to "genus Thauera" for proper English
A: Ok, done (line 298).
l 288: replace "at 75 nucleotides" by "minus 75 ..." or "75 nt 5' to..."
A: Ok, done (line 300).
l 291 in or to, but not into
A: Ok, done (line 302).
Fig. 2: the rbs indicated appears too long, Usually, only the longest stretch of complementary sequence to the 3’-end of 16S rRNA is taken into account (AGGAG in this case)
A: Ok, the RBS length has been modified in the Fig. 2 according to the reviewer suggestion.
L 332 and 412: replace thought by expected
A: Ok, done (lines 349 and 418)
l 369: "Azoarcus cells ... led to ... activity": poor English results in logical mess-up
Ok, done (lines 383-386).
l 386: using the ...plasmids... as DNA templates
A: Ok, done (line 407).
l. 390 slight
A: Ok, done (line 412).
l 393: at promoter PNI
A: Ok, done (line 415).
Fig. 5: this is my only substantial criticism: it is not sufficient to just show the behaviour of the one construct where nothing happens. This kind of experiment has to be acompanied by a control with the full promoter where the addition of B-CoA releases the binding. The authors can not refer to a previous paper here, but have to show that the constructs behave differently under identical experimental conditions (possible day-by-day variations may be caused e.g. by different degradation grades of protein or different hydrolysis ratios of B-CoA).
A: A new panel A has been included in Figure 5, to show the experiment with the PN promoter.
l 423: capital S for Svedberg constant
A: Ok, done (line 452).
l 430: do you mean disprove rather than discard?
A: OK, done (this sentence has been removed according the recommendation of Reviewer 1).
l 493: the dimeric protein BzdR
A. Ok, done (line 524).
Round 2
Reviewer 1 Report
The manuscript was considerably improved by the changes applied. A few comments are left (see below).
Comments
Line 65: ‘BzdY, 6-oxocyclohex-1-ene-1-carbonyl-CoA hydrolase’
Line 101: The ‘glycerol’ is missing in the current version of the manuscript.
Line 126: Apr (the ‘r’ is not superscript)
Line 198: ‘The presence of DNA was checked …’
Line 205: remove the ‘5’ at the end of this line
Line 229: The EcoRI site is not underlined.
Line 231: ‘Then, the DNA fragments …’
Line 289: The word ‘transcriptional’ should be removed as done in the introduction section.
Line 303/304: The wrong sequence was modified. Please check. It should be ‘TGCA(C)T(G/C)(C/G)A’ and ‘TGCA(N15)TGCA’.
Line 333. ‘with AcpR, a transcriptional activator from …’?
Line 364/365: ‘(C) Activity of the PN, PNII and PNI promoters’?
My old comment (Line 356/361: There is no detailed description available in the Material and Methods section. Please state the reference.), referred to the assay of β-galactosidase and not to the cultivation conditions. Sorry for the ambiguity. Please include in the figure legend a reference for this assay.
Line 377: ‘… generating the plasmids pBBR5-PN …’
Figure 4: The order of the panels for PNI and PNII should be changed as done for the columns in Fig. 3. In your answer you stated that this was done. In the current version, the original and not the revised figure is shown. The size of 184 nt cannot be deduced from the figure since no marker is shown.
In your answer to reviewers, you explained why NBzdRL in place of BzdR was applied in sedimentation equilibrium experiments. Possibly, this information should be open for the readership and therefore included in the main document (e.g. in the Material and Methods section).
Line 435/436: The binding of NBzdRL to PN was previously shown. PNI was firstly studied here.
Text related to Figure 7: Include the length of the DNA template without BzdR, which was analyzed by the same method (400 nm ±13 nm; PhD thesis Gutiérrez-Arroyo).
Author Response
Find enclosed the answer point-by-point to all the questions raised by the reviewer.
Yours sincerely,
Manuel Carmona
The manuscript was considerably improved by the changes applied. A few comments are left (see below).
Comments
Line 65: ‘BzdY, 6-oxocyclohex-1-ene-1-carbonyl-CoA hydrolase’
Ok, done (line 66).
Line 101: The ‘glycerol’ is missing in the current version of the manuscript.
Yes, we did not use glycerol in the set of experiment presented here.
Line 126: Apr (the ‘r’ is not superscript)
Ok, done (line 127).
Line 198: ‘The presence of DNA was checked …’
Ok, done (line 201).
Line 205: remove the ‘5’ at the end of this line
Ok, done (line 209).
Line 229: The EcoRI site is not underlined.
Ok, done (line 232).
Line 231: ‘Then, the DNA fragments …’
Ok, done (line 234).
Line 289: The word ‘transcriptional’ should be removed as done in the introduction section.
Ok, done (line 293).
Line 303/304: The wrong sequence was modified. Please check. It should be ‘TGCA(C)T(G/C)(C/G)A’ and ‘TGCA(N15)TGCA’.
Ok, done (lines 306/307).
Line 333. ‘with AcpR, a transcriptional activator from …’?
Ok, done (lines 336).
Line 364/365: ‘(C) Activity of the PN, PNII and PNI promoters’?
My old comment (Line 356/361: There is no detailed description available in the Material and Methods section. Please state the reference.), referred to the assay of β-galactosidase and not to the cultivation conditions. Sorry for the ambiguity. Please include in the figure legend a reference for this assay.
Ok, done (legend of Figure 3, lines 366 and 368).
Line 377: ‘… generating the plasmids pBBR5-PN …’
Ok, done (line 380).
Figure 4: The order of the panels for PNI and PNII should be changed as done for the columns in Fig. 3. In your answer you stated that this was done. In the current version, the original and not the revised figure is shown. The size of 184 nt cannot be deduced from the figure since no marker is shown.
The reviewer is right, sorry it was a mistake in figure processing. The panels of Figure 4 have been changed.
In your answer to reviewers, you explained why NBzdRL in place of BzdR was applied in sedimentation equilibrium experiments. Possibly, this information should be open for the readership and therefore included in the main document (e.g. in the Material and Methods section).
Ok, we have introduced this information in the Material and Methods section (lines 199-201).
Line 435/436: The binding of NBzdRL to PN was previously shown. PNI was firstly studied here.
Ok, the sentence “and PNI promoter” has been deleted, line 441.
Text related to Figure 7: Include the length of the DNA template without BzdR, which was analyzed by the same method (400 nm ±13 nm; PhD thesis Gutiérrez-Arroyo).
Ok, the sentence was included (lines 491-492).
Reviewer 2 Report
all my points have been met by the authors
Author Response
I would like to thank you the revision of this manuscript.
Your sincerely,
Manuel Carmona